# Factors Influencing Community Pharmacists’ Participation in Antimicrobial Stewardship: A Qualitative Inquiry

**DOI:** 10.3390/pharmacy13020056

**Published:** 2025-04-14

**Authors:** Tasneem Rizvi, Syed Tabish R. Zaidi, Mackenzie Williams, Angus Thompson, Gregory M. Peterson

**Affiliations:** School of Pharmacy and Pharmacology, University of Tasmania, Hobart, TAS 7005, Australia; tasneem.rizvi@utas.edu.au (T.R.); angus.thompson@utas.edu.au (A.T.); g.peterson@utas.edu.au (G.M.P.)

**Keywords:** antimicrobial resistance, antimicrobial stewardship, community pharmacists, qualitative study, antibiotics, Australia

## Abstract

Very few studies, all employing surveys, have investigated the perceptions of community pharmacists regarding antimicrobial stewardship (AMS). A qualitative inquiry exploring factors affecting community pharmacists’ participation in AMS may assist in the implementation of AMS in the primary care setting. This study aimed to explore the perceived barriers and enablers of community pharmacists’ participation in AMS. One-on-one semi-structured telephone interviews were conducted with a sample of community pharmacists from across Australia. Interviews were transcribed verbatim and analysed using the Framework Analysis method. Twenty community pharmacists (70% female), representing urban, regional, and remote areas of Australia participated in the study. Pharmacists identified a discord between clinical needs of patients and practice policies as the primary source of excessive prescribing and dispensing of antibiotics. The fragmented nature of the primary healthcare system in Australia was seen as limiting information exchange between community pharmacists and general practitioners about antibiotic use, that was encouraging inappropriate and, at times, unsupervised use of antibiotics. The existing community pharmacy funding model in Australia, where individual pharmacists do not benefit from any financial incentives associated with clinical interventions, was also discouraging their participation in AMS. Pharmacists suggested restricting default antibiotic repeat supplies, reducing legal validity of antibiotic prescriptions to less than the current 12 months, and adopting a treatment duration-based approach to antibiotic prescribing instead of the ‘quantity-based’ approach, where the quantity prescribed is linked to the available pack size of the antibiotic. Structural changes in the way antibiotics are prescribed, dispensed, and funded in the Australian primary care setting are urgently needed to discourage their misuse by the public. Modifications to the current funding model for pharmacist-led cognitive services are needed to motivate pharmacists to participate in AMS initiatives.

## 1. Introduction

Antimicrobial stewardship (AMS) is a coordinated set of interventions directed towards maximising the benefit of antimicrobial treatment while minimising harm, including the development of antimicrobial resistance [1]. The majority of AMS initiatives are developed and implemented in hospitals, although they are equally important to the primary care setting, where the majority of antibiotic use occurs [2]. Globally, it is estimated that up to 50% of antibiotic prescriptions in primary care are unnecessary [3]. Australia has one of the highest rates of community antibiotic use in the world [4] and is more than double that reported for some countries (e.g., The Netherlands), yet there is no AMS framework for the primary care sector in Australia [5].

The role of community pharmacists, unlike hospital pharmacists, within AMS is under-recognised in Australia and globally, despite their essential role in supporting the quality use of medicines and medicines optimisation [6]. Community pharmacists are often the first point of contact for the public seeking selfcare advice and symptomatic relief from minor ailments, such as self-limiting respiratory tract, urinary tract, eye, skin, soft tissue, and vulvovaginal infections [7]. In other situations, for instance where a patient already has an antibiotic prescription, the community pharmacist may also be the last point of professional contact and have an opportunity to promote appropriate antibiotic use [8].

Despite the current and potential role of community pharmacists as antimicrobial stewards, there has been little research exploring determinants of their involvement in AMS [9,10,11]. Our earlier investigations identified barriers and facilitators to Australian community pharmacists’ participation in AMS [10,12]. However, the inherent limitations of surveys prevented us from developing a deeper understanding of these issues within the context of clinical practice. This qualitative study was therefore undertaken to develop an in-depth understanding of the perceived barriers and enablers of community pharmacists’ participation in AMS, and to identify potential strategies to facilitate greater involvement of community pharmacists in AMS initiatives.

## 2. Materials and Methods

Community pharmacists who participated in an earlier national survey [10] were invited to participate in this follow-up qualitative study. Sixty participants expressed their interest by providing their email address, with ten responding to the subsequent invitation email. We recruited an additional ten Australian community pharmacists through a professional community pharmacists’ group with the social media platform, Facebook^®^. All participants were given an information sheet and provided informed consent. Participants were offered a gift voucher worth AUD 20 as a token of acknowledgement for their time.

An interview guide based on the findings of our national survey [10] was used to direct the interviews (Appendix A). Tasneem Rizvi (TR) piloted and modified the interview guide by conducting six mock interviews with the guidance of Angus Thompson (AT), three research pharmacists and three community pharmacists, before conducting the first interview. All interviews were conducted by TR, using a smartphone with a built-in audio recorder. The interviews were transcribed verbatim, and transcripts were verified by the participants before data analysis.

Interview transcripts were analysed using the Framework Method [13,14]. The first interview was analysed by all the team members and the team discussed any differences in the coding approach to reach an agreement. Following that, four additional interviews were analysed and reviewed together, and all the codes and disagreements were discussed. Based on the first five interviews, the initial thematic framework was developed. The final thematic framework was applied to the remaining interviews while accommodating additional codes or themes. No new codes emerged after the sixteenth interview. NVivo V.12 (QSR International, now known as Lumivero; Burlington, MA, USA) was employed for coding and organising the qualitative data. Ethics approval was obtained from the Human Research Ethics Committee of the University of Tasmania (H0015673). Gregory M Peterson (GP) reviewed the draft manuscript and provided expert feedback.

## 3. Results

The majority of the participants were female (70%) and represented a range of early, mid-career, and senior pharmacists representing urban, regional, and remote areas of Australia (Table 1). The interviews ranged from 16 to 43 min (mean: 29 min) in duration.

The perspectives of the participants indicated that community pharmacists are facing several challenges that limit their participation in AMS. Some key themes and sub-themes emerged from the interviews (Table 2).

### 3.1. Clinical and Practice Paradox

Many participants raised concerns about the way antibiotics are prescribed, dispensed, and consumed in the Australian primary care setting, that may be collectively grouped under the ‘clinical and practice paradox’. We defined this term as ‘clinical misuse of antibiotics due to practice-related limitations’. Participants not only shared the problems arising from this ‘clinical and practice paradox’, but also suggested solutions to overcome such problems.

In Australia, a prescription can be issued with ‘repeats’ (or ‘refills’) specified, i.e., the same prescription may be used for a further supply of medication without consulting with the original prescriber. Participants believed that the repeatable antibiotic prescriptions are encouraging the misuse of antibiotics in the community, particularly through enabling patients to self-medicate with antibiotics for future episodes of a ‘similar illness’. Pharmacists reported cases where patients had presented repeat antibiotic prescriptions for dispensing, several months after the original prescription (Table 2, 1.1). The participants reported that repeat prescriptions were often generated because the settings in electronic prescribing software automatically defaulted to issuing a repeat unless the general practitioner (GP) chose not to issue one. A few participants suggested a mandated change in the software default setting for antibiotic prescribing to zero repeats, to avoid the unintentional generation of automatic repeats.

Pharmacists reported that the available pack sizes of some antibiotics under the government-subsidised Pharmaceutical Benefits Scheme (PBS) in Australia are inconsistent with the recommended durations of antibiotic treatment in national antibiotic guidelines [15]. Participants stated that the discrepancy between the recommended duration and the pack size was often giving patients access to a surplus of antibiotics, encouraging overuse and self-medication in the future (Table 2, point 1.2). Some participants suggested that patients should be prescribed and given antibiotic quantities for an infection in accordance with clinical needs rather than the available pack size.

A third issue raised was the default legal validity of antibiotic prescriptions. Once written, antibiotic prescriptions in Australia remain valid for 12 months. This duration may be appropriate for chronic medications such as antihypertensive or anti-diabetic agents; however, having the same validity period for antibiotics provides patients with an ‘open prescription’ to be potentially (mis)used whenever they feel like obtaining it (Table 2, 1.3).

### 3.2. Fragmented Healthcare System

We define ‘fragmented healthcare’ as ‘a healthcare system where one healthcare provider is unaware of all episodes of patient care, involving a range of other healthcare providers. Many statements made by the participants pointed towards the lack of communication between multiple care providers and a disconnect between general practice and community pharmacy. The participants mentioned several clinical care issues that arose because GPs and pharmacists were essentially working in their “separate silos”.

In Australia, patients are not required to register with a general practice, and some will visit multiple practices and consequently multiple prescribers. Patients are also free to visit multiple community pharmacies with their prescriptions. Participants reported that the lack of a reliable and complete prescribing and dispensing history for all patients was contributing to antibiotic misuse (Table 2, point 2). Participants noted that some patients attend a different doctor to obtain antibiotic prescriptions if they are unsatisfied with the outcome of their initial consultation. Several suggestions were made by the participants to overcome problems arising from the fragmented healthcare system. These included the introduction of an antibiotic surveillance programme akin to that in place in Australia for pseudoephedrine. Such a system would enable monitoring and help ensure the dispensing of each antibiotic is recorded in a sharable database.

Similarly, the siloed nature of primary healthcare in terms of community pharmacists generally lacking access to the patient’s medical history and laboratory data was noted as another common limitation to community pharmacists supporting AMS (Table 2, point 2.1). One specific example where a lack of coordination between community pharmacists and GPs was evident was the way that ‘delayed prescribing’ is being implemented in Australian primary care. The participants reported that although they are increasingly seeing the trend of delayed prescribing being applied with antibiotics, it was generally in low-risk situations. In such cases, patients would sometimes still ask the pharmacist to dispense the antibiotic immediately (Table 2, point 2.2).

### 3.3. Nature of Community Pharmacy Practice

Several comments made by the participants were related to the nature and scope of community pharmacy practice in Australia. Community pharmacy runs as a business, and community pharmacists are largely reimbursed for activities on a fee for service model, whether these are dispensing or professional services.

The participants felt that the existing funding model in Australia favours community pharmacy owners, as the financial benefits of any government-funded clinical services within the pharmacy are provided to the pharmacy owners instead of employee pharmacists. Moreover, the proliferation of discount chains in the Australian community pharmacy sector, and declines in funding for dispensing services, were reported as adding pressure to the quality of care provided by community pharmacists (Table 2, point 3.1). Some pharmacists were also fearful of losing ‘customers’ to other pharmacies if they tried to educate patients or contact prescribers (Table 2, point 3.1).

After waiting to see and then consulting their GP, many patients, particularly when feeling unwell, expect that the prescribed antibiotic will be dispensed in the shortest possible time. Participants reported that, due to this patient pressure, they were usually unable to check details, such as whether the indication or duration was appropriate or not (Table 2, point 3.2). The participants also nominated ‘lack of time’ as one of the main reasons for not approaching the GP, despite identifying an obvious need to seek further clarification or intervening on a prescription.

### 3.4. Knowledge Base for Antimicrobial Prescribing

In Australia, Therapeutic Guidelines (TGs) are widely recognised to be the definitive source of information on appropriate antimicrobial prescribing [15]. However, participants were concerned that some GPs were not adhering to the recommendations in the TGs when prescribing antibiotics (Table 2, point 4). Furthermore, some participants also commented that not all pharmacists have access to TG. According to the participants, the lack of concordance to guidelines may lead to serious issues, particularly with children’s antibiotic doses.

### 3.5. Patients’ Understanding and Behaviours

Participants reported that patients lack an understanding of the consequences, implications, and effects of antimicrobial resistance. They stressed the need for better public understanding of the concept that the antibiotics will become ineffective if used unnecessarily. Participants pointed out the need for simpler and accessible messages to create more awareness (Table 2, point 5).

## 4. Discussion

The present study provides useful insights into the role of community pharmacists as antimicrobial stewards. Our participants noted system-wide issues that are contributing towards inappropriate antibiotic use, making these findings highly relevant to the broader healthcare community and organisations that are interested in implementing AMS in primary care.

We identified several issues affecting the prescribing and dispensing of antibiotics that are related to the wider healthcare system in Australia. Firstly, we noted the clinical intention of treating an episode of infection was implemented in an illogical manner to meet the restrictions imposed by the way medicines are funded through the PBS. The PBS dictates the quantity of each prescription (including a repeat/refill authorisation) under a particular prescribing and dispensing code. The prescribing and dispensing software in primary care, as well as the commercial packaging of antibiotics from manufacturers, are all designed to issue quantities aligned to the PBS schedule and not with treating common bacterial infections. The community pharmacist also faces barriers to developing their AMS role due to the effects of computer-generated antibiotic repeat prescriptions. The issue of repeat prescriptions was also highlighted in other Australian studies as a potential contributor to antibiotic misuse and overuse [16,17,18].

Secondly, the current legal validity of antibiotic prescriptions is another major potential source of inappropriate antibiotic use by the public. We found that patients were keeping their prescription for future use and may then have this dispensed several months after the date of issue. A study reported that one in ten antibiotics was dispensed from prescriptions that were more than one month old, although they were intended for the short-term treatment of acute infections [18]. The majority of acute infections in the community setting can be treated with a single 3-day or 7-day course of antibiotic, and a short legal validity of antibiotic prescriptions (e.g., one month) would reduce the chances of ill-informed self-medication.

Thirdly, we found community pharmacists’ inability to routinely access medical and pathology data was an important limiting factor in carrying out their AMS role. Integrating community pharmacies with the broader health system was strongly suggested to overcome this issue. This issue of fragmented healthcare systems has been reported in other studies that require system-wide changes to include community pharmacists in the primary healthcare team [19,20,21].

A qualitative study of 16 community pharmacists in France reported similar barriers to community pharmacists’ participation in AMS, such as difficult interactions with prescribers, lack of time, and lack of access to patient medical records and diagnoses. Strategies suggested by their participants to facilitate the implementation of AMS interventions included improving pharmacist–GP collaboration and financial incentives [22].

Unlike hospital pharmacy services, where AMS has become an essential component of pharmacists’ clinical role, community pharmacies are small businesses where the primary aim of their owners could be to generate profits. Pharmacists in our study reported that their skills, knowledge, and clinical capabilities were not currently being utilised as businesses were typically more focused on generating income-related activities, and there was a lack of any specific funding for AMS initiatives. Pharmacists also felt that the current funding arrangements only favours pharmacy owners and any limited incentives from clinical interventions are not passed on to the pharmacists conducting such interventions.

Pharmacists often have to engage with patients who seek antibiotics for wrong reasons—for viral infections, for self-medication, or for the treatment of other minor self-limiting illnesses. This illustrates poor public awareness of the consequences of inappropriate antibiotic use and of antimicrobial resistance as a threat to public health. Our findings are in line with other studies conducted in Australia, where limited health literacy regarding antimicrobial resistance was identified despite mass education campaigns [23,24]. The participants suggested smart, short, and simple messages to improve patients’ understanding of bacterial and viral infections, antimicrobial resistance, self-medication, and minor self-limiting illnesses.

Despite the challenges outlined above, it was encouraging to see a general willingness and enthusiasm from community pharmacists about their role in community-based AMS initiatives. Provided that adequate resources and support are available, pharmacists were keen to participate in educating patients, collaborating with GPs and triaging patients with respiratory symptoms to discourage antibiotic use for viral infections. Educational interventions to facilitate short consultations and counselling with patients by the community pharmacists through information leaflets have been successful in European countries and may be applied in the Australian context [21,25].

To the best of our understanding, this is the first qualitative study of Australian community pharmacists that explored specific barriers and facilitators to their participation in AMS initiatives. Our participants represented various regions of Australia, ranging from remote regional areas to major metropolitan centres. Likewise, a range of age and experience was represented. This was a single country study and therefore the views expressed by the Australian community pharmacists may not represent the global community pharmacists’ perspective. Similarly to any cross-sectional study, our study may also have had a selection bias as the participating community pharmacists may have had a desire to become involved in AMS.

While the authors acknowledge the value of numerical representation for clarity, the qualitative nature of this study data provides a comprehensive understanding of the factors influencing community pharmacists’ participation in AMS. It is suggested that future studies consider not only identifying and highlighting key themes but also the frequency of responses, while maintaining the integrity of the qualitative insights for detailed data representation.

## 5. Conclusions

Health system-related issues, role-based limitations, interprofessional dynamics and resource constraints continue to limit the ability of Australian community pharmacists to participate in AMS. Community pharmacists can play an important role in AMS, particularly when advising patients. However, a number of changes in health policy and practice are required to facilitate this process.

## Figures and Tables

**Table 1 pharmacy-13-00056-t001:** Demographic Profile of Study Participants.

#	State/Territory	Sex	Experience (in years)	Area	Pharmacy Size
1	NSW	F	6	Suburban	Medium
2	NSW	M	8	Suburban	Small
3	SA	F	7	Rural	Large
4	NSW	F	11	Regional	Medium
5	NSW	M	30	Rural	Small
6	VIC	M	9	Metropolitan	Large
7	VIC	F	3	Suburban	Medium
8	WA	F	24	Suburban	Medium
9	TAS	F	4	Rural	Medium
10	WA	F	1	Suburban	Large
11	WA	F	9	Metro	Large
12	QLD	F	4	Metro	Large
13	WA	F	9	Suburban	Small
14	ACT	F	3	Suburban	Medium
15	NSW	M	10	Suburban	Large
16	QLD	F	8	Suburban	Medium
17	QLD	F	10	Regional	Large
18	QLD	M	2.5	Urban	Medium
19	QLD	M	16	Metro	Medium
20	TAS	F	3	Suburban	Large

**Table 2 pharmacy-13-00056-t002:** Themes, Sub Themes and Related Codes.

Themes and Sub-Themes	Representative Quotes
Clinical and practice paradox	
1.1 Repeat authorisation	“They are getting access to something they should not have access to, that is getting a repeat.” “GP’s computer just automatically selects a repeat, even if [they] said that ‘it is for five days’, em-----? they just give it to the patient.” “Get rid of that automatic repeat or put an expiry on repeat prescriptions.”
1.2 Pharmaceutical Benefits Scheme quantity	“A very common example is a treatment for UTI: trimethoprim they say it is for three days. But it comes in packs of seven.” “We know that they do not need the balance of the medication and obviously at the end of the day, what they do with it we do not know. “We have to have a course duration specified for all antibiotic treatments for patients, and we only give them what they need for that course.”
1.3 Validity of antibiotic prescriptions	“A lot of the time the patient just grabs an antibiotic script from like half a year ago and then when you ask them reason what that is for? Then you start to realise that it might not actually be the right antibiotic for that type of infection.”
2. Fragmented healthcare system	“--in community pharmacy, it is much harder to implement those kinds of systems collaboratively with GPs, where we are working in different settings.” “If people travel from one pharmacy to another, you then have to rely on what the patient tells you.” “They have a script for erythromycin from one GP. They were not happy with it, saw another doctor, pretended that they have not seen a doctor and that doctor prescribed them Augmentin Duo Forte. Then they come and see you and they say ‘oh, I have these two, what is the difference, which is better?”
2.1 Inadequate information	“In terms of knowledge, we definitely have it------but we do not have access to full patient’s medical records.”
2.2 Delayed prescribing	“You can feel that they’ve been pressured and bullied by the customer, because now a lot of doctors’ scripts will have statements like ‘withhold for 72 h, do not treat unless fever above 38.5C’. And then the patients come in the same day, saying ‘the doctor said not to take it, but I’d rather take it anyway”
3. Nature of community pharmacy practice	
3.1 Funding model	“There is no push in doing so. So, whether you do that or whether you do not [reference-AMS] that it is alright yeah. It’s like still there is no initiative for you to do that [AMS].” “If we tell them that no, this is not the right antibiotic. You are sending away business in the commercial pharmacy setting.”
3.2 Time pressures of being the last stop on patients’ journey	“It is not all the time I get the doctor immediately, so if the doctor is busy and then the patient is in a hurry, we usually end up with what has been prescribed, which is not appropriate.”
4. Knowledge base for antimicrobial prescribing	“All the GPs that I’ve called use old resources, released once and never updated.” “They don’t weigh the child and they just prescribe the dose based on the standard. So, I do not know which guideline they use to write, but not based on the child’s weight.”
5. Patients’ understanding and behaviours	“Lots of patients are saying, ‘No, no, there is just too much information’. They want something in a sentence or two. They don’t want a leaflet/pamphlet about resistance.”

## Data Availability

The original contributions presented in this study are included in the article. Further inquiries can be directed to the corresponding author(s).

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
