# Peer review of "Factors Influencing Community Pharmacists’ Participation in Antimicrobial Stewardship: A Qualitative Inquiry"

_pharmacy, 2025, doi:10.3390/pharmacy13020056_

Round 1
Reviewer 1 Report
Comments and Suggestions for Authors
The study is highly relevant to Antimicrobial Stewardship (AMS) initiatives, providing key insights into how community pharmacists can contribute to the fight against antimicrobial resistance. It underscores the importance of improving public education, enhancing collaboration between GPs and pharmacists, and fostering a more integrated healthcare system to support AMS in primary care. The study has several strengths, including a diverse range of participants and a structured methodology. The research aim is clearly defined, and the findings are clearly presented but some limitations should be considered when interpreting the findings.
- Small sample size: Only 20 participants were involved in the study, which limits the depth of understanding and the transferability of the findings. Could the authors justify the sample size based on the research goals and approach?
- The study acknowledges the possibility of selection bias, as the community pharmacists who participated may have had a particular interest in AMS, as they had already participated in a national survey and expressed interest in the follow-up which could have influenced their responses. How would the authors overcome this limitation?
- While the study covered different regions (urban, regional, and remote areas), it does not mention how these regions were represented within the study. If most of the participants came from suburban areas, for instance, the findings may not fully reflect the challenges faced by community pharmacists in other areas.
Reviewer 2 Report
Comments and Suggestions for Authors
Thanks for giving me a chance to review. I read with great interest your study on qualitative research within community pharmacists in Australia and voicing their concerns on antibiotic stewardship. I liked your paper but here are some ways to make it better:
- I like the fact that your respondents were a good mix of both urban and rural pharmacists.
- I know you had a reasonable sample size but were older pharmacists who are less likely to be familiar with modern technology (a generalization) facing more issues with the user interface that hampers their ability to provide proper advise.
- CDC has technological features to prevent opioid abuse. https://www.cdc.gov/overdose-prevention/hcp/clinical-guidance/prescription-drug-monitoring-programs.html Does Australia have something similar to Antibiotic tracking?
- Will using Antibiotic tracking systems help address some of the concerns raised by pharmacists?
- Also your study was conducted in a high resource country like Australia. Do you have any research from developing countries?
Reviewer 3 Report
Comments and Suggestions for Authors
After critically reviewing this Research Article titled "Factors Influencing Community Pharmacists’ Participation in Antimicrobial Stewardship: A Qualitative Inquiry", I detected some MINOR flaws, which determined my recommendation of “ACCEPT UNDER MINOR REVIEW”. Below please find my detailed comments.
The researchers worked on a qualitative study that explored the factors that affect community pharmacists’ participation in antimicrobial stewardship (AMS) in a personal way, through telephone calls and recorded interviews. The aim of the research was to raise relevant issues that could assist in the implementation of AMS in the primary care context.
The idea behind this work is quite interesting and could, in the long term, improve the health system in that region of Australia, the subject of this article. As mentioned, this is a situation that applies specifically to some regions of the world, since the laws for prescribing and modifying prescriptions for medications for patients vary from country to country. For example, in many countries medical and/or dental prescriptions for antibiotics are valid for a maximum of 30 days after the appointment.
The authors highlighted very well the key points of the problem in the country of research, which would be the very long period of validity of the prescription, quantities dispensed to patients that do not correspond to the ideal treatment time for that illness and prescriptions that can be reused without a new appointment with the doctor and/or dentist. All of this leads to an extremely worrying scenario of self-medication.
My suggestions are bellow:
- Although this is qualitative research, this reviewer believes that transforming such data into a numerical table, for example, in each of the questions in the questionnaire, how many percent of the interviewees responded in such a way, would bring greater credibility and a more tangible visualization of the results obtained.
